# Network Approaches to Integrate Analyses of Genetics and Metabolomics Data with Applications to Fetal Programming Studies

**DOI:** 10.3390/metabo12060512

**Published:** 2022-06-02

**Authors:** Alan Kuang, M. Geoffrey Hayes, Marie-France Hivert, Raji Balasubramanian, William L. Lowe, Denise M. Scholtens

**Affiliations:** 1Department of Preventive Medicine (Biostatistics), Northwestern University Feinberg School of Medicine, 680 N. Lake Shore Drive, Suite 1400, Chicago, IL 60611, USA; alan.kuang@northwestern.edu; 2Department of Medicine, Northwestern University Feinberg School of Medicine, Rubloff 12, 420 E. Superior St, Chicago, IL 60611, USA; ghayes@northwestern.edu (M.G.H.); wlowe@northwestern.edu (W.L.L.J.); 3Department of Population Medicine, Harvard Pilgrim Health Care Institute, Harvard Medical School, Boston, MA 02215, USA; mhivert@partners.org; 4Diabetes Unit, Endocrine Division, Massachusetts General Hospital, Boston, MA 02114, USA; 5Department of Medicine, Université de Sherbrooke, Sherbrooke, QC J1K 2R1, Canada; 6Department of Biostatistics and Epidemiology, University of Massachusetts-Amherst, Amherst, MA 01003, USA; rbalasub@schoolph.umass.edu

**Keywords:** fetal programming, variant-to-metabolite associations, Bayesian network modeling, serial mediation modeling, integrated multi-omics

## Abstract

The integration of genetics and metabolomics data demands careful accounting of complex dependencies, particularly when modelling familial omics data, e.g., to study fetal programming of related maternal–offspring phenotypes. Efforts to identify genetically determined metabotypes using classic genome wide association approaches have proven useful for characterizing complex disease, but conclusions are often limited to a series of variant–metabolite associations. We adapt Bayesian network models to integrate metabotypes with maternal–offspring genetic dependencies and metabolic profile correlations in order to investigate mechanisms underlying maternal–offspring phenotypic associations. Using data from the multiethnic Hyperglycemia and Adverse Pregnancy Outcome (HAPO) study, we demonstrate that the strategic specification of ordered dependencies, pre-filtering of candidate metabotypes, incorporation of metabolite dependencies, and penalized network estimation methods clarify potential mechanisms for fetal programming of newborn adiposity and metabolic outcomes. The exploration of Bayesian network growth over a range of penalty parameters, coupled with interactive plotting, facilitate the interpretation of network edges. These methods are broadly applicable to integration of diverse omics data for related individuals.

## 1. Introduction

Fetal programming describes the series of fetal responses to in utero stimuli, including the maternal genetic and metabolic milieu, during critical phases of cellular differentiation that may substantially impact fetal development [1]. According to the Barker hypothesis, these programmed adaptations may be the origin of a number of diseases in later life [2]. Associations between maternal metabolic phenotypes, including glycemia and body mass index, with newborn adiposity are well documented, e.g., in the Hyperglycemia and Adverse Pregnancy Outcome (HAPO) study findings [3,4]. Related investigations using high-dimensional omics data for HAPO mothers and their offspring have provided some clarity around the mechanism of these associations, and include investigations relating maternal genetics and maternal metabolic phenotypes [5], maternal metabolomics and maternal metabolic phenotypes [6], maternal metabolomics and newborn outcomes [7], fetal genetics and newborn outcomes [8,9], and cord blood metabolomics and newborn outcomes [10]. These analyses have tended to focus on pairs of maternal or newborn predictors and outcomes in HAPO and other cohorts, with only limited attempts at integrating multi-omics data [11].

To provide a unified framework for more fully interrogating multi-omics fetal programming contributions that underlie maternal–offspring phenotype associations, we adapt a Bayesian network modeling approach. A Bayesian network is a graphical probabilistic model defined from a set of variables and their conditional dependencies via a directed acyclic graph (DAG). This analytic strategy has been successfully deployed for other analyses of omics data [12,13,14,15,16]. The unique idea behind Bayesian networks is to estimate a series of directed relationships among nodes in which the distribution of a variable represented by a ‘child’ node is described conditional on its ‘parent’ nodes, with a penalty term ‘λ’ typically applied to induce model sparsity [17]. Ordering of relationships is particularly relevant in multi-omics data analyses. For example, genotypes can affect metabolite levels, but metabolite levels do not affect genotype. In maternal–offspring studies, maternal genotype affects offspring genotype, but not vice versa. Implicit ordering of maternal/newborn metabolite associations is also often identifiable. Directionality of possible associations may be explicitly defined for Bayesian network models, thus accommodating the natural ordering of maternal–offspring and genetic-metabolomics data. Using this cohesive Bayesian network approach, we identify overall trends in the relative contribution of genetics and metabolomics to maternal–offspring phenotypic associations. Additionally, we uncover potential ordered multi-step mediation pathways that are independently validated. This comprehensive multi-omics analytic pipeline for correlated maternal–offspring data provides an alternative to common analytic strategies that emphasize direct associations between omics data sources and phenotypes.

Interpretation and utilization of this scale of integrated omics analysis demands sophisticated graphical display that is accessible to multidisciplinary teams of investigators. To complement Bayesian network estimation, we apply visualization techniques that demonstrate network growth over a range of penalty parameters for the model. In so doing, we observe the varied strengths of multi-omic associations with the HAPO maternal–offspring phenotypes of interest. The approach to comprehensive network visualization is complemented by the ability to highlight specific ordered paths of interest when undertaking strategic analysis of potential mediation mechanisms.

## 2. Results

### 2.1. Significant Variant-to-Metabolite Associations

Across 5.2 million genotyped and imputed SNPs and 331 metabolites (43 unique fasting and 1-hr metabolites from a 2-hr 75g OGTT conducted during HAPO at ~28 weeks’ gestation and/or metabolites from cord blood collected at delivery), a total of 510 unique SNPs had *p* < 10^−7.5^ in per-metabolite GWAS analyses. Forty-three pairs of maternal and newborn SNPs remained after LD trimming (Appendix A). Notable identified associations included *APOA5* (rs2072560 and rs2266788) with fasting maternal triglycerides, *GCKR* with fasting and 1-hr maternal 2-hydroxybutyric acid and maternal 1-hr lactate, and *SLC16A9* (rs1171616, rs1171617, and rs1171619) with fasting and 1-hr maternal AC C2 (Figure 1). The strongest association found was *CPS1* (rs1047891 and another SNP 2500 base-pairs away) with fasting and 1-hr maternal glycine. Three newborn SNPs (rs7601356, rs2286963, and rs3764913) in the *ACADL* region and one newborn SNP (rs3738934) in the *RPE* region were highly associated with cord blood AC C8:1-OH/C6:1-DC. Forty-nine newborn SNPs in the *REV3L* region were highly associated with cord blood NEFA. The 43 pairs of maternal and newborn SNPs listed in Appendix A were included as variables for estimating Bayesian network models (BNMs). All significant GWAS results are listed in Appendix A.

### 2.2. Bayesian Network Model (BNM)

Under the highest penalty term value of λ = 15 (i.e., the most stringent penalty parameter and thus the sparsest network), the BNM involving fasting maternal glucose and metabolites estimated edges among a network involving 187 nodes with at least one incident edge and 137 edges. As λ values decreased, the number of nodes in each estimated BNM with at least one incident edge and the total number of edges increased, culminating in a BNM involving 309 nodes and 3070 edges for the lowest value of λ = 1 (i.e., the least stringent penalty parameter and thus the fullest network). Similarly, for the BNM involving 1-hr metabolites and 1-hr maternal glucose, the most stringent penalty of λ = 15 resulted in a BNM with 180 nodes and 132 edges and grew to 306 nodes and 3033 edges at λ = 1. Appendix A demonstrate network growth for networks that were estimating using fasting and 1-hr maternal metabolites, and Table 1 catalogs the numbers of edges connecting nodes of different types at various values of λ. 

Dynamic animation of both the fasting and 1hr BNMs were generated to better visualize areas of increasing edge density over the range of penalty parameters (Appendix A). Nodes were colored to indicate whether they represent phenotypes, genotypes, and metabolites, with different colors for each metabolite compound class. Node shapes were used to indicate whether they represent maternal or offspring variables. As evidenced in Table 1 and in the Appendix A video animation, most BNM edges exist among maternal metabolites and cord blood metabolites under the most stringent penalty, with edges primarily existing within metabolites of the same class. Connections increase between the maternal and cord blood metabolites before maternal exposure phenotypes and newborn outcome phenotypes connect to the BNM at λ = 7 for both the maternal fasting and 1-hr BNMs. While maternal–offspring genotypes are pairwise connected by edges under the most stringent penalty as would be expected, the first pairwise edge between a genotype node to a non-genotype node appears at λ = 10 for the fasting BNM and λ = 9 for the 1-hr BNM, where in both instances rs715 from *CPS1* connects to glycine. The *CPS1*-glycine association is well-documented in multiple GWAS studies [18,19,20,21]. Substantial connections from genotype nodes to the larger network do not occur until far less stringent λ values are used BNM. This network growth phenomenon suggests that fetal programming mechanisms are likely to be more substantially informed by environmentally driven metabolic contributions than strictly inherited germline genetic variation that may predispose an individual to certain phenotypes.

### 2.3. Serial Mediation Model (SMM)

For the fasting and 1-hr BNMs with the least stringent penalty values at λ = 1, all shortest paths from a maternal feature to a newborn outcome (sum of skinfolds (SSF), birthweight, cord C-peptide or cord glucose) were identified. Any path that was a subpath of a larger path was not considered. In total, 3000 unique directed paths from a maternal feature to a newborn outcome were identified in the fasting BNM and 1438 unique paths were identified in the 1-hr BNM. The number of nodes ranged from 3 to 8 in the paths extracted from both the fasting and 1-hr BNMs.

For all identified 4438 different pathways, SMM analyses were performed as described. We specifically examined pathways for which the first node was treated as the independent ‘exposure’ variable and the last node, a newborn outcome (i.e., SSF, birthweight, cord glucose or cord C-peptide), was the dependent ‘outcome’ variable, and all nodes in between were treated as potential mediators, i.e., individual mediators for paths with three nodes or serial mediators for paths with four or more nodes. For the 3000 candidate pathways from the fasting BNM, nine demonstrated a statistically significant IDE after false discovery rate (FDR) adjustment and maintained a positive proportion mediated (PM) in the training data set. For the 1438 candidate pathways from the 1-hr BNM, 22 demonstrated a statistically significant indirect effects (IDE) after FDR adjustment and maintained a positive PM in the training data set. When SMM analyses were repeated in the testing dataset, statistically significant serial mediation was validated for five of these pathways (Figure 2 and Figure 3, Table 2). Notably, only one of the pathways has a genetic component, consistent with previous observations of the stronger metabolomics dependencies in this HAPO data set.

## 3. Discussion

Integrated modeling of multi-omics data demands careful specification of complex dependencies. For HAPO genetics and metabolomics data, the expected dependencies between observations for mother–offspring pairs merits additional consideration, particularly given the potential for investigating mechanisms underlying fetal programming of related clinical phenotypes in mothers and their newborns using these unique data resources. Recent efforts to identify ‘genetically determined metabotypes’ using classic GWAS approaches in independent individuals identified SNP-metabolite associations with larger effect sizes than typically observed for clinical phenotypes [19,20,22,23,24]. Extensions to GWAS, such as phenotype set enrichment analyses to identify SNPs associated with multiple metabolites [25] and informatics resources to investigate purported metabotypes [26,27,28], have also proven useful for probing the genetics and metabolomics of complex disease. However, to accurately characterize genetic and metabolic contributions to fetal development, more sophisticated models are required to integrate metabotypes with maternal–fetal genetic dependencies and metabolic profile correlations. BNMs offer a natural paradigm for this type of data integration due to the ability to naturally incorporate structured, ordered dependencies among different types of data while estimating edges.

A key finding upon global review of our integrated analyses is the relative strength of coordinated behavior among classes of maternal and newborn metabolites compared to all other possible dependencies among the variables included in the estimated BNMs. While we may expect high dependency within maternal and offspring metabolites, the relative density of edges connecting maternal and offspring metabolites under the strictest penalty parameters highlights the tight coordination between the metabolisms of a mother and the developing fetus. Of note, network edges between maternal and offspring metabolites and the maternal and offspring clinical phenotypes emerge prior to any edges between the paired clinical phenotypes (e.g., maternal glucose to cord blood glucose) and shared maternal/offspring genetics. This observation suggests that environmentally derived or influenced metabolomic contributions may outweigh the genetic underpinnings of fetal programming for size and adiposity at birth.

Complementing the big picture linking of metabolomics and genetics contributions to maternal/offspring phenotypes is the specific pathway-level interpretation of possible mediating mechanisms. Serial mediation analyses informed by comprehensive BNM estimation indicated biologically plausible mediation mechanisms. In the pathway from 1-hr tyrosine to cord glucose via cord blood tyrosine, phenylalanine and propionyl carnitine (C3), the BNM suggests that higher levels of maternal 1-hr tyrosine indirectly result in lower cord glucose and revealed the plausible intermediate pathways linking these two. It is known that phenylalanine is an essential amino acid and is metabolized to tyrosine when the tyrosine level is low. However, if maternal tyrosine is high and is transported across the placenta leading to sufficient fetal tyrosine, then the conversion of phenylalanine to tyrosine is reduced. C3 is formed from amino acid metabolism but primarily branch chain amino acids (BCAA). Previous studies have shown that the aromatic acids, phenylalanine, tyrosine, and the BCAA, are associated with insulin resistance in adults, including in pregnancy [11]. Thus, it is plausible that they could be involved in fetal glycemic regulation. For the pathway from *SLC16A9* to cord C-peptide via cord blood acetylcarnitine (C2) and hexadecenoylcarnitine (C16:1), *SLC16A9* is a transporter of monocarboxylates that has been associated with C2 levels [29]. Elevated C2 levels in the mother may affect C2 in the fetus through placental transfer. While the direct relationship between C2 and C16:1 is unclear, both C2 and C16:1 levels are higher in newborns with extreme macrosomia [24] and higher levels of C16:1 have been noted in newborns of mothers with GDM [30]. Notably, the biological plausibility of these purported SMM pathways is based on observations in adult metabolism. Our findings when jointly examining maternal and newborn data with the described dependencies suggest similar pathway involvement very early in life. Furthermore, examination of serial pathways in the BNM highlights that while maternal glucose is typically emphasized when studying newborn adiposity and glucose/insulin regulation, in fact multiple maternal metabolic factors, including metabolites that may cross the placenta, are part of the fuller picture.

Penalty parameter selection for sparse estimation of edges in large scale networks is a topic of focused discussion and ongoing bioinformatics research. In contrast to selecting a single penalty parameter that optimizes specific criteria for estimation of one BNM of our integrated data, we opted to view and animate BNMs over a range of penalty parameters. This approach to visualization, along with color distinctions for node types, naturally facilitated the observation of the relative centrality of maternal/offspring metabolite dependencies compared to shared genetics for explaining dependencies between clinical phenotypes that were specifically investigated in HAPO. For other studies involving multi-omics data, exploring network growth over decreasing stringency requirements may illuminate the relative strengths and cooperativity of different types of omics features. 

There are limitations to our analytic approach. In our BNM analyses, we blocklisted potential edges from offspring SNPs to maternal phenotypes and fasting/1-hr maternal metabolites such that no such edge could be included in the estimated models. However, there is evidence that offspring genotype can have limited influence maternal phenotypes [31,32]. Implementing the specified blocklist prevented the discovery of such rare occurrences with our data. We chose this condition to reduce noise from these edges for BNM estimation. A second limitation pertains to the examination of serial mediation in the BNM as independent paths. As shown, there are two purported mediation pathways from maternal 1-hr tyrosine to cord glucose. Another approach on this specific pathway would be to use a combination of both parallel and serial mediation modeling. In addition, we focused only on serial mediation paths that started at a maternal variable and ended at a newborn outcome. Other routes of investigation for different combinations of pathway starting and end points may yield further insight about mediating mechanisms. 

In summary, Bayesian network modeling and animated visualization over a range of penalty parameters provided a natural analytic framework for the cogent synthesis of genetics and metabolomics data related to clinical phenotypes for HAPO mothers and their newborns. This approach facilitated both large-scale summaries of the relative contributions of data types, as well as local interrogation of potential serial mediating pathways. These pathways may suggest unique lines for further laboratory and/or population-level investigation, potentially incorporating other edges from the BNM that influence the linear serial mediating pathway of primary interest. Beyond the HAPO study, the approaches described here can be adapted to integrate omics data for other studies that are designed to evaluate complex contributions that may underlie clinical phenotypic associations.

## 4. Materials and Methods

### 4.1. Data Sets

#### 4.1.1. Hyperglycemia and Adverse Pregnancy Outcome Study

The HAPO Study was an observational, population-based study conducted from 2000–2006 at 15 international field centers that recruited over 25,000 participants [3]. Enrolled women underwent a 75-g oral glucose tolerance test (OGTT) at approximately 28 weeks’ gestation after an overnight fast. Fasting, 1-hr, and 2-hr glucose levels were measured as described [33,34]. Maternal blood samples during the OGTT were placed on ice immediately, processed within 60 min of collection, stored at −20 °C or −80 °C for 1–6 weeks, shipped on dry ice to the HAPO Central Laboratory, and stored at −80 °C. Each participant’s age, gestational age at time of OGTT, family history of diabetes and hypertension, parity, and history of cigarette smoking and alcohol use were obtained by questionnaire. Cord blood samples were obtained within 5 min of delivery and before delivery of the placenta, processed within 30 min, and stored at −70 °C until laboratory assays were performed. Cord blood C-peptide, birthweight, and sum of skinfolds (SSF) were measured as described using calibrated equipment and standardized methods [33,34,35]. The HAPO protocol was approved by each field center’s institutional review board, and participants gave informed consent.

#### 4.1.2. Metabolomics Data

Conventional clinical and targeted metabolites were measured in 3463 maternal fasting and 1-hr serum samples [36] and in 1600 offspring cord blood serum samples [10,37]. Conventional clinical metabolites (triglycerides, nonesterified fatty acid (NEFA), lactate, glycerol, and 3-hydroxybutyrate) were measured using standard enzymatic chemistries on a Beckman-Coulter Unicell DxC 600 clinical analyzer. Targeted panels of amino acids and acylcarnitines were analyzed by flow-injection, electrospray-ionization tandem mass spectrometry (MS) and quantified by isotope or pseudo-isotope dilution using a Waters TQ triple quadrupole MS, equipped with an Acquity liquid chromatography system, and with data handling in the MassLynx 4.1 environment (Waters Corporation, Milford, MA, USA). Five conventional clinical metabolites, 15 amino acids, and 45 acylcarnitines were analyzed across maternal fasting, 1-hr, and offspring cord blood. Targeted metabolites with >10% missing data across all samples were excluded from analysis, yielding a total of 57, 56, and 48 targeted metabolites for analysis in maternal fasting, 1-hr, and offspring cord blood, respectively.

Non-targeted assays were performed using gas chromatography-mass spectrometry (GC-MS) as described [38]. Briefly, maternal fasting, 1-hr, and offspring cord blood serum samples were extracted with methanol that was spiked with a retention time-locking internal standard of perdeuterated myristic acid. After methanol extraction, drying and derivatization by methoximation and trimethylsilylation, samples were run in daily batches of matched sets of fasting and 1-hr maternal OGTT sera and cord blood sera on a 7890B GC/5977B MS (Agilent Technologies, Santa Clara, CA, USA). Each batch also included multiple injections of a quality control (QC) sample and a process blank. QC samples consisted of uniform pools generated from aliquots of all sera in the study and were injected at the beginning, middle and end of each batch. GC-MS peaks were deconvoluted with AMDIS freeware [39] (http://www.amdis.net/ accessed on 22 June 2021 [40]) and annotated using the Agilent Fiehn GC-MS Metabolomics RTL Library [41] with additions from the laboratory at Duke University School of Medicine. Only annotated peaks were included for analysis. Integrated peak areas were log_2_ transformed. QC data were then used to account for technical variability on a feature-specific basis, with batch correction and normalization performed as described using the *metabomxtr* R package (version 1.26.0) [42,43]. Non-targeted metabolites with >20% missing data across all samples were excluded from analysis. Remaining non-targeted metabolites with <20% missing data were imputed using metabolite-specific minimum values since, in general, non-targeted metabolite missing values are attributable to low abundance. Non-targeted metabolite measures that were also represented in the targeted assays were excluded from analysis. In total, 55, 53, and 47 non-targeted metabolites were included in the final analyses from the maternal fasting, 1-hr, and offspring cord blood samples, respectively.

#### 4.1.3. Genotype Data

Genotyping and QC for HAPO data have been described for an initial set of 2581 maternal and newborn Afro-Caribbean and 1615 maternal and newborn Mexican American DNA samples (Illumina HumanOmni1M-Duo v3B SNP array, Illumina, San Diego, CA, USA), 3152 maternal and newborn Northern European samples (Illumina Human610-Quad v1B SNP array, run at the Broad Institute, Cambridge, MA, USA), and 2466 maternal and newborn Thai samples (Illumina HumanOmni-Quad v1-0B SNP array, run at the Center for Inherited Disease Research, Baltimore, MD, USA) [5,9].

In addition, a subset of 2680 Northern European maternal and newborn samples were run using the Illumina Human 610 Quad v1 B SNP array at the Broad Institute Center for Genotyping and Analysis (CGA), USA [44], and a total of 5564 maternal and newborn transethnic samples were processed on the Illumina Global Screening Array-24 v2.0 A1 following agreed-upon protocols of the Gene, Environment Associations Studies consortium [45]. Exclusion criteria for samples and SNPs included inconsistent self-report of sex at birth with available chromosomal data, chromosomal anomalies, unintended sample duplicates, sample relatedness based on genetic relatedness, low call rate (samples with missing call rate >1% and SNPs with missing call rate >2%), SNPs with >3 Mendelian errors, departures from Hardy–Weinberg equilibrium (< 1 × 10^−4^ for the original Afro-Caribbean, Mexican American, Northern European, and Thai groups, <1 × 10^−6^ for the additional 2680 Northern European group, and <1 × 10^−3^ for the transethnic group), duplicate discordance, sex differences in heterozygosity, low minor allele frequencies (<0.01) and/or low imputation quality score (<0.75).

Genotype imputation was performed separately on the twelve datasets (six maternal and six fetal) on the TOPMed Imputation Server using Minimac4 (version 1.5.7) [46] and the TOPMed reference panel for the Afro-Caribbean, Mexican American, Northern European, Thai, Northern European and transethnic samples [47]. Consistent strand assignments between the reference dataset and the QC-cleaned and filtered datasets were ensured using the strand-checking utility of McCarthy Group Tools. Strand was corrected and/or SNPs were removed where strandedness could not be resolved. The HAPO genotype was then imputed to the TOPMed reference panel, using an r^2^ filter of 0.3 to remove unreliably imputed SNPs. A total of 5,237,318 overlapping SNPs were analyzed in a mega-analysis across the twelve imputed datasets. Principle components (PCs) were estimated separately for maternal and offspring data using the package *SNPRelate* version 1.26.0 in R [48].

#### 4.1.4. Combined Genotype, Metabolomics and Phenotype Training and Validation Data Sets

For analyses reported here, 1385 HAPO mother/offspring pairs with all of the following data types were retained for analysis: (1) maternal genotype, (2) offspring genotype, (3) maternal fasting and 1-hr targeted and non-targeted metabolomics, (4) offspring cord blood targeted and non-targeted metabolomics, (5) maternal fasting and 1-hr glucose measurements and maternal BMI from the HAPO OGTT at ~28 weeks’ gestation along with additional maternal data used for model adjustments, and (6) offspring sum of skinfolds, birthweight, cord glucose, and cord C-peptide measured at delivery. Two-thirds of these samples (*n* = 924) were used as a training data set for Bayesian network model development and selection of potential mediation pathways. The remaining one-third (*n* = 461) were used as a validation data set for confirming statistical significance of purported mediation findings. Sample selection for training and validation data sets was stratified by ancestry group based on genotype data.

### 4.2. Statistical Analyses

#### 4.2.1. Variant-to-Metabolite Associations

In order to reduce computational burden when estimating Bayesian networks, an initial round of ‘metabotyping’ was performed, i.e., genome wide association (GWAS) analyses were performed separately for all metabolites as outcomes [22,26]. Metabotyping was performed on the combined training and validation data sets using a mega-analysis linear regression approach with SNPTest v2.5 against an additive genotype variable, separately for maternal fasting and 1-hr metabolites with maternal genotype, and offspring cord blood metabolites with offspring genotype. Maternal fasting metabolite models were adjusted for the first 3 principal components of genetic ancestry, maternal age, gestational age, BMI, height, parity (0/1+), mean arterial pressure and fasting plasma glucose (1-hr glucose for 1-hr metabolites) at HAPO OGTT, and sample storage time. Newborn cord blood metabolites were adjusted for the first 3 principal components of genetic ancestry, maternal age, BMI, height, parity (0/1+), mean arterial pressure at OGTT, gestational age at delivery, newborn sex, and sample storage time. SNPs with nominal *p* < 10^−7.5^ were selected for further analysis. This selection threshold was set to be slightly more inclusive than typical genome-wide significance of nominal *p* < 10^−8^, anticipating that the penalized network modeling approach would aid in the selection the most informative SNPs. Each metabolite with significant SNP associations was LD trimmed independently using the package *SNPRelate* version 1.26.0 in R [48].

#### 4.2.2. Bayesian Network Modeling (BNM)

We estimated BNMs including the following variables as nodes in the network: maternal exposure variables, namely BMI at OGTT, log_10_ fasting plasma glucose at OGTT, square root of 1-hr plasma glucose at OGTT (fasting and 1-hr glucose transformations were consistent with previous GWAS analyses of maternal phenotypes [5]), the 43 SNPs identified in preliminary metabotyping, 117 fasting maternal metabolites, 114 1-hr maternal metabolites, 100 cord blood metabolites, and a set of newborn anthropometric outcomes (birthweight, square root sum of skinfold, log_10_ cord glucose, and log_10_ cord C-peptide; data transformations were consistent with previous GWAS analyses of these newborn phenotypes [9]). A simple schematic illustrating the types of ordered relationships that might be estimated among maternal and fetal genotypes and metabolites and a corresponding set of conditional dependencies that are central to BNM estimation are illustrated in Figure 4. BNMs were estimated using the training data set, as specified above. We used a fast, score-based method that uses sparse regularization and block-cyclic coordinate descent approach to build these BNMs [17]. Conditions on edge directionality were specified using a ‘blocklist’, with the following edge types omitted from the range of potential estimated edges: offspring SNPs to maternal SNPs; newborn outcomes to maternal SNPs; cord blood metabolites to maternal SNPs; offspring SNPs to maternal phenotypes; offspring SNPs to fasting/1-hr maternal metabolites; newborn outcomes to offspring SNP; cord blood metabolites to offspring SNPs; maternal phenotypes to offspring SNPs; fasting/1-hr maternal metabolites to maternal SNPs; fasting/1-hr maternal metabolites to offspring SNPs; and SNPs from one chromosome to SNPs in different chromosomes. A vector of λ values was used in the solution path, starting at λ = 15 and decreasing to λ = 1, and networks were estimated across a range of these regularization penalty parameters. Separate BNMs were estimated for fasting maternal metabolites/fasting maternal glucose and for their 1-hr counterparts. Since metabolism at the fasting and 1-hr post-glucose timepoints may reflect different components of glucose metabolism, we chose to model data from the fasting and 1-hr timepoints separately.

#### 4.2.3. Serial Mediation Model (SMM)

Shortest directed paths in the estimated Bayesian network under the lowest λ value (i.e., least stringent penalty parameter) were identified as possible candidates for mediation analyses using shortest path identification functionality in the *igraph* R package [49]. In particular, we focused on shortest directed paths connecting a maternal SNP, fasting/1-hr metabolite, or maternal phenotype as a starting point along a directed path to a newborn outcome as the end point of the path. These paths were interpreted as purported mediation pathways connecting a maternal exposure to a newborn phenotype and thus potentially informative for investigating mechanisms of fetal programming. All paths containing 3 or more nodes were examined using serial mediation model (SMM) analysis with structural equation modeling using the *lavaan* R package version 0.6-11 [50]. The serial mediator Model 6 from Model Templates for PROCESS for SPSS and SAS [51] was adapted to test statistical significance of serial mediation through different pathways involving maternal genetics, maternal fasting/1-hr metabolomics, maternal phenotypes, offspring genetics, and cord blood metabolomics to inform a newborn outcome. Indirect effects (IDE), direct effects (DE), and proportion mediated (PM) were estimated and standard errors were estimated through bootstrapping. False discovery rate adjustment was applied to IDE p-values separately for the fasting BNM and 1-hr BNM paths [52]. SMMs were adjusted for the following covariates at the HAPO OGTT: field center, maternal age, gestational age, height, parity (0/1+), mean arterial pressure, sample storage time, offspring sex, and gestational age at delivery. SMM analysis was on performed on the training dataset. Those pathways with significant indirect effects Benjamini-Hochberg adjusted *p*-values < 0.05) and a positive proportion mediated were evaluated for replication in the testing dataset according to nominal *p*-value < 0.05.

#### 4.2.4. Network Visualization

We generated interactive visualizations to assist in the interpretation of BNMs over a range of penalty parameters using functions from the *igraph* version 1.2.4.1 [49], *intergraph* version 2.0-2 [53], *networkDynamic* version 0.10.0 [54], *network* version 1.15 [55,56], and *ndtv* version 0.13.0 [57] R packages. Graphs were animated over 15 penalty terms starting at λ = 15 and decreasing to λ = 1 to demonstrate network growth as the penalty decreased. The *x*-axis on the.html files that contain the network animations index the increasing density of the BNM estimates, i.e., an index of 1 on the *x*-axis in the animation corresponds to λ = 15, an index of 2 to λ = 14, and so on.

## Figures and Tables

**Figure 1 metabolites-12-00512-f001:**
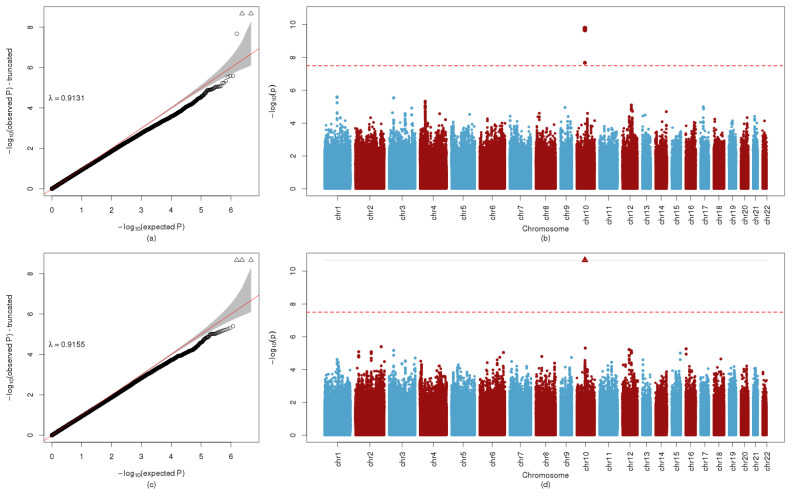
QQ and Manhattan Plot AC C2. (**a**) QQ plot of GWA with fasting AC C2, with corresponding genomic inflation factor, λ. (**b**) Manhattan plot of GWA with fasting AC C2. The red line indicates -log_10_p = 7.5, the criterion used for potential inclusion in the BNM. (**c**) QQ plot of GWA with 1-hr AC C2, with corresponding genomic inflation factor, λ. (**d**) Manhattan plot of GWA with 1-hr AC C2. The red line indicates −log_10_p = 7.5, the criterion used for potential inclusion in the BNM.

**Figure 2 metabolites-12-00512-f002:**
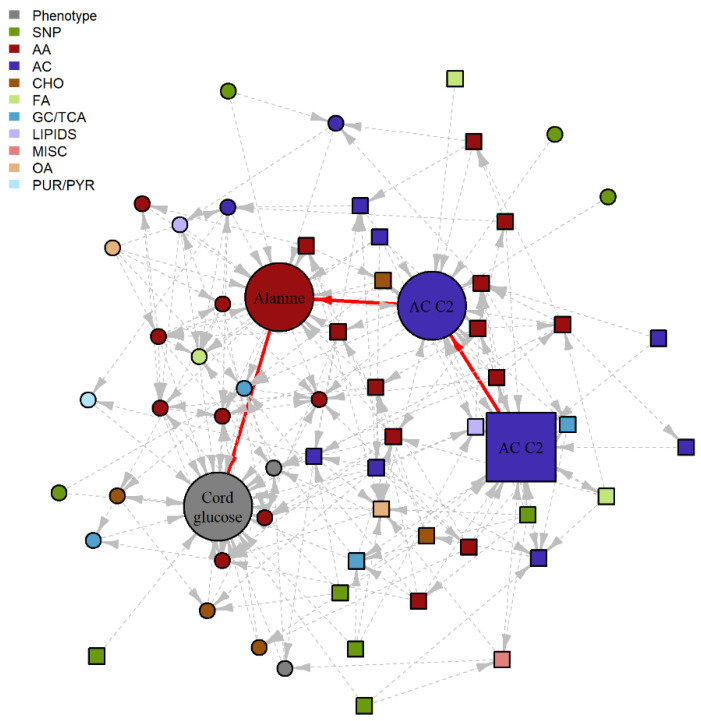
Subgraph of fasting BNM highlighting a potential SMM pathway. Nodes represent the metabolite, genotype and phenotype variables and edges represent the conditional dependencies between them. The maternal features are represented by square nodes and the offspring features are represented by circle nodes. Newborn outcome variables are grey, genotypes are green, and the remaining colors correspond to different metabolite classes. The solid, red edges correspond to the purported SMM pathway with statistically significant IDE after FDR correction and positive proportion mediated in the training data set, with nominal significance and positive PM confirmed in the validation data set. Dashed, grey edges correspond to nodes adjacent to the SMM pathway. AA, amino acid; AC, acylcarnitine; CHO, carbohydrate; FA, fatty acid; GC/TCA, glycolysis/tricarboxylic acid cycle; OA, organic acid; PUR/PYR, purine or pyrimidine.

**Figure 3 metabolites-12-00512-f003:**
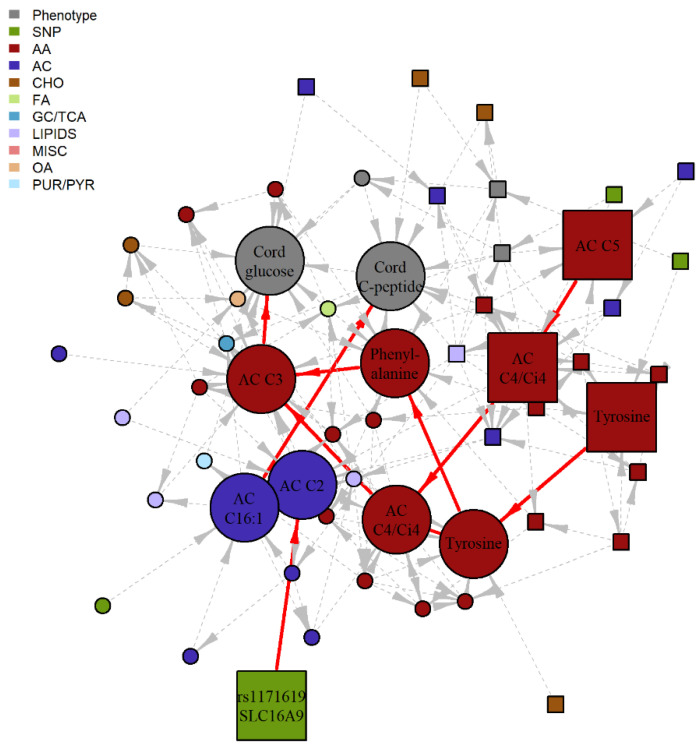
Subgraph of 1-hr BNM highlighting four potential SMM pathway. Nodes represent the metabolite, genotype and phenotype variables and edges represent the conditional dependencies between them. The maternal features are represented by square nodes and the offspring features are represented by circle nodes. Newborn outcome variables are grey, genotypes are green, and the remaining colors correspond to different metabolite classes. The solid, red edges correspond to the four purported SMM pathways involving 1-hr metabolites with statistically significant IDE after FDR correction and positive proportion mediated in the training data set, with nominal significance and positive PM confirmed in the validation data set. Dashed, grey edges correspond to nodes adjacent to the SMM pathways. AA, amino acid; AC, acylcarnitine; CHO, carbohydrate; FA, fatty acid; GC/TCA, glycolysis/tricarboxylic acid cycle; OA, organic acid; PUR/PYR, purine or pyrimidine.

**Figure 4 metabolites-12-00512-f004:**
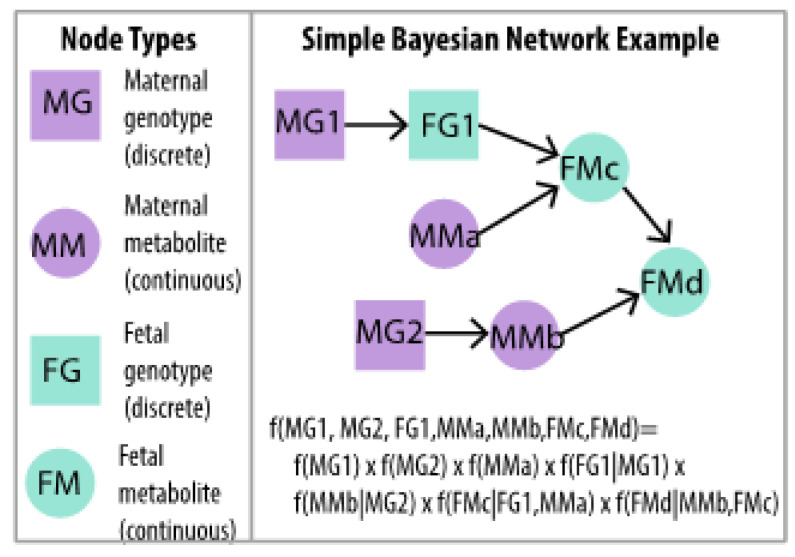
Simple BNM example of two maternal genotypes (MG1, MG2), one fetal genotype at the same locus as one maternal genotype (FG1), two maternal metabolites (MMa, MMb) and two other fetal metabolites (FMc, FMd) in which the full multivariate distribution is written as a product of conditional distributions. Ordered dependencies among maternal and fetal genotypes and metabolites are naturally accommodated in BNM estimation.

**Table 1 metabolites-12-00512-t001:** Numbers of edges among types of nodes in estimated Bayesian Network Models (BNMs) for a range of λ values.

**BNMs Involving Fasting Metabolites and Fasting Maternal Glucose**					
	Numbers of edges—N (% of total)				
Types of nodes in each edge pair	λ = 15	λ = 10	λ = 7	λ = 2	λ = 1
MG-MG	8 (6%)	9 (4%)	11 (3%)	16 (1%)	16 (1%)
MG-MM	0 (0%)	1 (0%)	3 (1%)	293 (10%)	292 (10%)
MG-MP	0 (0%)	0 (0%)	0 (0%)	8 (0%)	22 (1%)
MG-NG	26 (19%)	37 (15%)	38 (9%)	45 (2%)	45 (1%)
MG-NM	0 (0%)	0 (0%)	0 (0%)	114 (4%)	113 (4%)
MG-NP	0 (0%)	0 (0%)	0 (0%)	8 (0%)	8 (0%)
MM-MM	49 (36%)	84 (34%)	159 (37%)	904 (31%)	901 (29%)
MM-MP	0 (0%)	0 (0%)	3 (1%)	8 (0%)	20 (1%)
MM-NM	4 (3%)	18 (7%)	44 (10%)	629 (22%)	627 (20%)
MM-NP	0 (0%)	0 (0%)	0 (0%)	30 (1%)	44 (1%)
MP-MM	0 (0%)	0 (0%)	1 (0%)	31 (1%)	92 (3%)
MP-MP	0 (0%)	0 (0%)	1 (0%)	1 (0%)	1 (0%)
MP-NM	0 (0%)	0 (0%)	0 (0%)	13 (0%)	60 (2%)
MP-NP	0 (0%)	0 (0%)	2 (0%)	4 (0%)	5 (0%)
NG-NG	8 (6%)	9 (4%)	11 (3%)	11 (0%)	11 (0%)
NG-NM	0 (0%)	0 (0%)	0 (0%)	95 (3%)	95 (3%)
NG-NP	0 (0%)	0 (0%)	1 (0%)	6 (0%)	6 (0%)
NM-NM	41 (30%)	90 (36%)	155 (36%)	662 (23%)	661 (22%)
NM-NP	0 (0%)	1 (0%)	1 (0%)	16 (1%)	16 (1%)
NP-NM	0 (0%)	0 (0%)	0 (0%)	30 (1%)	30 (1%)
NP-NP	1 (0%)	1 (0%)	1 (0%)	4 (0%)	5 (0%)
λ = 15 most stringent penalty, λ = 1 least stringent penalty λ = 10 first occurrence of genotype to non-genotype edge λ = 7 first occurrence of maternal exposure phenotype to offspring outcome phenotype λ = 2 first occurrence of SMM pathway evaluated from maternal fasting AC C2 to cord glucose outcome					
**BNMs involving 1-hr Metabolites and 1-hr Maternal Glucose**					
	Numbers of edges—N (% of total)				
Types of nodes in each edge pair	λ = 15	λ = 9	λ = 7	λ = 3	λ = 1
MG-MG	8 (6.06%)	10 (3.38%)	11 (2.54%)	16 (1.08%)	16 (0.53%)
MG-MM	0 (0%)	1 (0.34%)	3 (0.69%)	89 (6.02%)	272 (8.97%)
MG-MP	0 (0%)	0 (0%)	0 (0%)	8 (0.54%)	34 (1.12%)
MG-NG	26 (19.7%)	37 (12.5%)	38 (8.78%)	45 (3.04%)	45 (1.48%)
MG-NM	0 (0%)	0 (0%)	0 (0%)	21 (1.42%)	93 (3.07%)
MG-NP	0 (0%)	0 (0%)	0 (0%)	1 (0.07%)	10 (0.33%)
MM-MM	43 (32.57%)	105 (35.47%)	156 (36.03%)	507 (34.28%)	893 (29.44%)
MM-MP	0 (0%)	1 (0.33%)	4 (0.92%)	12 (0.81%)	40 (1.32%)
MM-NM	5 (3.79%)	27 (9.12%)	41 (9.47%)	242 (16.36%)	628 (20.71%)
MM-NP	0 (0%)	0 (0%)	1 (0.23%)	14 (0.95%)	47 (1.55%)
MP-MM	0 (0%)	1 (0.34%)	4 (0.93%)	16 (1.08%)	59 (1.95%)
MP-MP	0 (0%)	0 (0%)	0 (0%)	0 (0%)	0 (0%)
MP-NM	0 (0%)	0 (0%)	0 (0%)	4 (0.27%)	55 (1.81%)
MP-NP	0 (0%)	0 (0%)	1 (0.23%)	4 (0.27%)	4 (0.13%)
NG-NG	8 (6.06%)	10 (3.38%)	11 (2.54%)	11 (0.74%)	11 (0.36%)
NG-NM	0 (0%)	0 (0%)	0 (0%)	23 (1.56%)	101 (3.33%)
NG-NP	0 (0%)	0 (0%)	1 (0.23%)	1 (0.07%)	8 (0.26%)
NM-NM	41 (31.06%)	102 (34.46%)	160 (36.95%)	442 (29.88%)	664 (21.89%)
NM-NP	0 (0%)	1 (0.34%)	1 (0.23%)	10 (0.68%)	18 (0.59%)
NP-NM	0 (0%)	0 (0%)	0 (0%)	9 (0.61%)	30 (0.99%)
NP-NP	1 (0.76%)	1 (0.34%)	1 (0.23%)	4 (0.27%)	5 (0.17%)
λ = 15 most stringent penalty, λ = 1 least stringent penalty λ = 9 first occurrence of genotype to non-genotype edge λ = 7 first occurrence of maternal exposure phenotype to offspring outcome phenotype λ = 3 first occurrence of all SMM pathways involving maternal 1-hr metabolites MG = maternal genotype MM = maternal metabolite MP = maternal phenotype NG = newborn genotype NM = newborn metabolite NP = newborn phenotype					

**Table 2 metabolites-12-00512-t002:** Individual ordered pathways from Bayesian Network Models (BNMs) demonstrating statistically significant serial mediation in training and validation data sets.

PATHWAY MEMBERS	Training BNM	Validation Data
	IDE (95% Confidence Interval); FDR-Adjusted P, PM (%)	IDE (95% Confidence Interval); Nominal P, PM (%)
Maternal rs1171619 *SLC16A9* -> Cord blood AC C2 -> Cord blood AC C16:1 -> Cord C-peptide	−1.60 × 10^−2^ (−2.42 × 10^−2^–−7.78 × 10^−3^); 0.0086, 25.51%	−1.36 × 10^−2^ (−2.41 × 10^−2^–−3.20 × 10^−3^); 0.0120, 173.98%
Maternal fasting AC C2 -> Cord blood AC C2 -> Cord blood Alanine -> Cord glucose	1.26 × 10^−2^ (6.12 × 10^−3^–1.90 × 10^−2^); 0.0202, 129.82%	1.21 × 10^−2^ (1.38 × 10^−3^–2.29 × 10^−2^); 0.0151, 24.17%
Maternal 1-hr AC C5 -> Maternal 1-hr AC C4/Ci4 -> Cord blood AC C4/Ci4 -> Cord blood AC C3 -> Cord glucose	−1.21 × 10^−2^ (−1.72 × 10^−2^–−6.94 × 10^−3^); 0.0018, 28.58%	−8.86 × 10^−3^ (−1.45 × 10^−2^–−3.23 × 10^−3^); 0.0101, 17.63%
Maternal 1-hr Tyrosine -> Cord blood Tyrosine -> Cord blood AC C4/Ci4 -> Cord blood AC C3 -> Cord glucose	−8.06 × 10^−3^ (−1.20 × 10^−2^–−4.13 × 10^−3^); 0.0053, 55.5%	−1.13 × 10^−2^ (−1.83 × 10^−2^–−4.35 × 10^−3^); 0.0067, 64.19%
Maternal 1-hr Tyrosine -> Cord blood Tyrosine -> Cord blood Phenylalanine -> Cord blood AC C3 -> Cord glucose	−7.55 × 10^−3^ (−1.16 × 10^−2^–−3.48 × 10^−3^); 0.0106, 52.02%	−8.04 × 10^−3^ (−1.41 × 10^−2^–−2.03 × 10^−3^); 0.0261, 45.56%

## Data Availability

The maternal and newborn genotype data on European ancestry, Mexican American, Thai and Afro-Caribbean proposed for use in these studies and the accompanying phenotype data are currently available through dbGaP (www.ncbi.nlm.nih.gov/gap, accessed on 1 June 2021). The remaining additional subset of 2680 Northern European and transethnic samples. metabolomic data and codes used for analyses will be made available by the authors upon request.

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
