# Peer review of "Network Approaches to Integrate Analyses of Genetics and Metabolomics Data with Applications to Fetal Programming Studies"

_metabolites, 2022, doi:10.3390/metabo12060512_

Round 1

Reviewer 1 Report

The manuscript entitled “Network Approaches to Integrate Analyses of Genetics and Metabolomics Data with Applications to Fetal Programming Studies” deals with the integration of multi-ethnic hyperglycemia and adverse pregnancy outcome genetics and metabolomics data integrated into a Bayesian network which elucidates the mechanisms for fetal programming of newborn adiposity and metabolic outcomes.

Overall, the manuscript is well written and is easy to understand. The references used in the manuscript are recent and are adequate.  Regarding the novelty of the manuscript, it provides new tools to decipher the mechanisms underlying metabolomics and genetics for size and adiposity at birth., and as far as I know this is the first study of its kind. The experiments carried out were enough and suitable for the purpose of the manuscript.

In my opinion, the results shown in this manuscript are interesting for a broader community.

Best regards

Author Response

We are grateful to this reviewer for the favorable reviews.

Reviewer 2 Report

This is brilliant work done by the authors and a well-detailed presentation of the methodology.

Kindly revisit the manuscript for some minor text errors (line 130, 264 etc) and also please have a second look at the figures and tables referenced in the text to make sure they are in the correct order (kindly consider removing from the supplementary material any parts that are preceded in the main text).

Also please revisit an empty page that seems to appear in the middle of the manuscript.

Author Response

We’re delighted that the reviewer is so pleased by the submission. We fixed the minor text error noted in line 130, as well as a few others throughout the submission. We reviewed line 264 (which in the original submission was the line where the Friedman et al. (2000) reference was listed) and we couldn’t find the text error. We also removed the empty page that must have resulted from reformatting after the submission upload. We’d be happy to do any final fixes with further guidance. We double checked the order of Figures and Tables in main text and, as far as we can tell, it appears the order is correct.

Regarding the supplemental files, we submitted very long tables that seemed inappropriate for the main manuscript (e.g., specific genetic variants identified in metabotyping studies) as well as animated .html files as supplemental material. If the editors would prefer that the longer tables be included in the main text, or if there is a mechanism for including the .html files in the main manuscript, we would be very glad to make further edits.

Reviewer 3 Report

The manuscript under the title “Network Approaches to Integrate Analyses of Genetics and  Metabolomics Data with Applications to Fetal Programming Studies” represents an interesting original scientific paper describing the application of Bayesian network models to 

integrate multi-omics (metabolomics and genotype data) fetal programming contributions that underlie maternal-offspring phenotype associations in the HAPO (Hyperglycemia and Adverse Pregnancy Outcome) study of mothers and their newborns.

The manuscript is well written with an appropriate introductory section, clearly described materials, and method section, and an appropriately explained and illustrated result section followed by a sufficient discussion subsection and a relevant list of up-to-date references covering the corresponding research field.

There is only a slight correction requested related to the titles of result subsections 3.2, 3.3 and headings of tables 1 and 2 – the incorporation of full names for Bayesian Network Modeling (BNM)  and Serial Mediation Model (SMM) is suggested with the corresponding changes of 3.2 and 3.3 result subsection titles if possible.

Other than that, the manuscript publication in its current form is suggested.

Author Response

Thank you for the positive reviews of the submitted work. We have made the subsection and table heading revisions as suggested.

Reviewer 4 Report

The manuscript is well-written and provides all background to the reader less familiar with the subject. The study relies on a sound experimental set-up and proves of high conceptual and practical skills of the investigators. The conclusions are valid and fully supported by the results.

Author Response

We are grateful for the positive comments on the submission.